# Interaction of $Co_3O_4$ Nanocube with Graphene and Reduced Graphene Oxide: Adhesion and Quantum Capacitance

**Vladislav Shunaev** [1,*] **and Olga Glukhova** [1,2]

1    Department of Physics, Saratov State University, 410012 Saratov, Russia; glukhovaoe@sgu.ru
2    Institute for Bionic Technologies and Engineering, Sechenov University, 119991 Moscow, Russia
*    Correspondence: shunaevvv@sgu.ru; Tel.: +7-8452-514562

**Abstract:** The composites on the base of $Co_3O_4$ and graphene are in demand in the field of portable, flexible energy storage devices due to their small size, lightweight, big specific capacitance, good cycle stability and appropriate capacitance retention. The synthesis of this material always starts from the treatment of graphene oxide, so as a result, experimenters receive $Co_3O_4$ nanocubes incorporated into reduced graphene oxide indicates the presence of different oxygen-containing groups in the compound. This fact may limit the advantages of the considered material. Our theoretical quantum chemical calculations show that the process of $Co_3O_4$ incorporation between reduced graphene oxide layers is more energetically favorable in comparison to pure graphene. However, the win in the quantum capacitance in the case of pure graphene is in the range of 300–500 F/g in dependence on the applied voltage. The obtained result may indicate the need for modification of the current methods of graphene/$Co_3O_4$ synthesis to improve its application in supercapacitors and lithium-ion batteries.

**Keywords:** graphene; $Co_3O_4$; lightweight supercapacitors; DFTB; heat of formation; adhesion; quantum capacitance





## 1. Introduction

Lightweight and flexible supercapacitors (SCs) with the planar and fiber-shaped structures are essential elements of portable and wearable devices—one of the most rapidly developing technologies right now [1,2]. Transition metal oxides are considered the most promising materials for chemical power sources such as supercapacitors (SCs) and lithium-ion batteries (LIBs) [3,4]. Among them, $Co_3O_4$ attracts increased attention due to its high theoretical capacitance (3560 F/g), low cost, ease of synthesis, large surface area and the best compromise between initial capacity and capacity retention [5–7]. However, its application is limited by extremely low electrical conductivity and drastic volume change during Li+ ion insertion/extraction leading to poor cyclability. The integration of $Co_3O_4$ with superconductive and ultimately strong graphene (G) solves this problem. The electrodes on the base of $Co_3O_4$/G demonstrate high-specific capacitance, fast electron transfer and long-cycle life [8–16]. Besides that, graphene and its derivatives have already been considered as an element base for fully flexible, lightweight, high-performance all-solid-state SCs. [17,18].

Li et al. reported on a prepared nanosheet assembly route-layered G/$Co_3O_4$ hybrid electrode that exhibited a high specific capacitance of 331 F/g at a current density of 5 A/g and good cycle stability with 122% capacitance retention after 5000 continuous charge–discharge cycles [8]. The $Co_3O_4$/G/$Co_3O_4$ sandwich architecture obtained by sonochemical synthesis demonstrated high specific capacitance of 276.6 F/g at a current density of 0.5 A/g and good cycling stability with 92.4% capacitance retention after 5000 cycles at 5 A/g [9]. Synthesized via a hydrothermal method and constant current electropolymerization, the sandwich-structured $Co_3O_4$@C@PPy electrode delivered a higher areal capacitance of 2.71 F/cm$^2$ at 10 mA/cm$^2$ (1663 F/g at 6.1 A/g) [10]. Synthesized by a facile hydrothermal process and subsequent thermal treatment, the sandwich-like $Co_3O4$/$_G$ nanocomposites

maintained a specific capacity of 639.8 mAh/g at a current density of 0.5 C after 50 cycles with a capacity retention rate of 71%. Cong et al. received a hierarchal porous structure that represented ultrasmall $Co_3O_4$ anchored into ionic liquid (IL)-modified graphene oxide (GO), ultrasmall/$Co_3O_4$-GO-IL electrodes were delivered by a remarkably simple hydrothermal and annealing process [11]. The obtained structure demonstrated electrochemical properties with an excellent initial capacity of 1304 mAh/g at a current density of 0.5 C and durable long-term cycling stability of capacity retention of 98.4% even after 500 cycles. Yang et al. reported a novel one-step laser irradiation method that synthesized 2D graphene sheets decorated by numerous ultrafine $Co_3O_4$ nanoparticles. This structure reached 978.1 F/g at the current densities of 1 A/g and showed 93.7% capacitance retention of 916.5 F/g at current densities up to 10 A/g [12].

Several attempts were performed to discover the energy and electron properties of $Co_3O_4$/G nanocomposite by density functional theory (DFT) methods. Feng et al. studied the coupling effect at the interface of graphene. It was found that the graphene sheet bent on the $Co_3O_4$ surface due to the strong electron coupling at the interface, and a Co-C bond with a length of 1.897 Å was formed [19]. Beatty et al. reported that the strong interaction of G with $Co_3O_4$ might be explained by the charge transfer while the binding energy was equal to 0.193 eV/atom [20]. Cong confirmed the significant charge transfer from G to the top surface of $Co_3O_4$ but noted that the minimal distance between G and $Co_3O_4$ was equal to 3.4 Å [11]. Odedairo reported that the shortest equilibrium distances between the graphene layer and the $Co_3O_4$ nanosheet were calculated to be 2.6 Å [21]. However, all these theoretical works considered the interaction of $Co_3O_4$ particles with pure graphene, while in all real experiments, the synthesis of GO was the essential initial stage [8–16]. In fact, the $Co_3O_4$/G nanocomposite represented the compound of $Co_3O_4$ and reduced graphene oxide (rGO); that is why the presence of oxygen atoms at the graphene surface should be taken into account. Besides that, recent studies have demonstrated the presence of in-plane oxygen in graphene [22,23]. In particular, the so-called "oxygen pair" and "graphitic oxygen" are two configurations where oxygen directly substitutes carbon atoms in the graphene lattice and constitute ~50% of the total relative counts. Substitutional oxygen in the graphene lattice could modify the adhesion parameters between $Co_3O_4$ and graphene, as well as the overall calculated capacitance of the nanocomposite, so graphene with in-plane oxygen should also be considered in this study.

This paper aims to theoretically find out the regularities of $Co_3O_4$ nanocubes binding to the surface of pure graphene, graphene with substitutional oxygen and graphene oxide with the following obtaining of $Co_3O_4$/rGO nanocomposite by the self-consistent charge density-functional tight-binding (SCC DFTB) method. The most favorable hybrid materials for lightweight SCs will be found from the view of energetic and capacitive parameters.

## 2. Methods

The optimization of the atomic structure, as well as the calculation of zone structure, was performed by the SCC DFTB method [24]. Besides the traditional DFTB terms, the Lennard-Jones potential [25] was introduced to evaluate the interaction of non-bonded atoms. All calculations were performed at the real temperature of 0 K; the SCC tolerance was $1 \times 10^{-5}$. The Brillion zone was sampled according to the Monkhorst-Pack scheme with parameters $8 \times 8 \times 1$ for 2D films and $8 \times 8 \times 8$ for 3D composites.

The quantum capacitance (QC) $C_Q$ was first introduced as a part of the total capacitance $C_T$ that required extra energy for filling a quantum well with electrons due to the Pauli principle [26]:

$$\frac{1}{C_T} = \frac{1}{C_G} + \frac{1}{C_Q} \tag{1}$$

For low-dimensional structures, QC was comparable to geometric capacitance $C_G$ and played a key role in processes of charge and discharge. In our research, $C_Q$ was calculated by the formula [27,28]:

$$C_Q = \frac{1}{mV} \int_0^V eD(E_F - eV')dV' \tag{2}$$

where $m$ is the mass structure, $V$ is the applied voltage that corresponds to shift in the Fermi level $E_F$, $D$ is the area under the plot of density of state (DOS) in the considered range and $e$ is the elementary charge.

## 3. Results

### 3.1. Adhesion

In the first stage, we attached $Co_3O_4$ nanocubes from the Fd3m space group [29] that was observed in [30] to 160 atoms of pure graphene. The most favorable structure was received when the atomic cell was optimized in the periodic box with lattice vectors Lx = 19.78 Å and Ly = 21.41 Å. Energy calculations showed that the complex was stabilized by 7.37 eV:

$$\Delta H (G \rightarrow G/Co_3O_4) = E(G/Co_3O_4) - E(G) - E(Co_3O_4) = -7.37 \text{ eV} \tag{3}$$

We observed the formation of the covalent bond between Co and C atoms with the length of 2.07 Å, which was a repeat of the results of Beatty et al. [20]. After binding, the $Co_3O_4$ nanocube reported the charge of 1.16 e to the graphene. We noticed that the $Co_3O_4$ particle maintained its cubic form in contrast to $Fe_3O_4$ particles that demonstrated partial loss of crystal after attachment to the graphene layer [31].

The atomic structure of the graphene with substitutional graphitic oxygen (G*) is shown in Figure S1. Incorporating into the graphene carcass, the oxygen atom formed two C–O bonds with a length of 1.47 Å and one C-O bond with a length of 1.48 Å. The C–C length of carbon atoms bonded to oxygen decreases to 1.40 Å. After the attachment of the $Co_3O_4$ nanocube, the C–O bond with a length of 1.48 Å was broken, and the C atom was bonded to the Co atom with a length of 2.09 Å (Figures 1b and S2). Another C–Co bond of 2.07 Å was observed, as in the case of the $G/Co_3O_4$ nanocomposite. The heat of the G*/Co3O4 formation was lower than for $G/Co_3O_4$ due to the presence of two chemical bonds and was equal to:

$$\Delta H (G^* \rightarrow G^*/Co_3O_4) = E(G^*/Co_3O_4) - E(G^*) - E(Co_3O_4) = -9.69 \text{ eV} \tag{4}$$

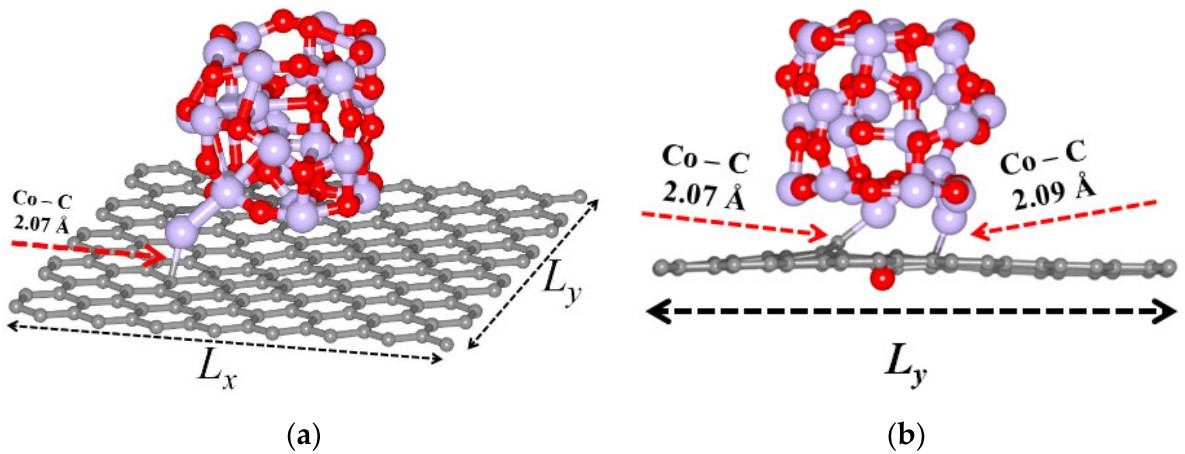

(**a**)          (**b**)

**Figure 1.** The atomic structure of (**a**) the $G/Co_3O_4$ nanocomposite and (**b**) the $G^*/Co_3O_4$ nanocomposite. The translation vectors are Lx = 19.78 Å and Ly = 21.41 Å.

As was mentioned in the Introduction, in the process of $G/Co_3O_4$ nanocomposite synthesis, $Co_3O_4$ nanoparticles are firstly disposed of at the GO surface, and then the

oxygen-containing groups are etched. The scheme of this process modeling is shown in Figure 2. In the first stage, the graphene particle was oxidized with the maximum possible energy epoxy functional groups. The number of oxygen atoms was equal to 17. Thus the received mass ratio in obtained GO was m(G):m(O) = 14.2%. Note that after optimization, the lattice vectors were enlarged and were equal to Lx = 19.90 Å, Ly = 21.62 Å. The Co–C bond that was observed in the case of the $G/Co_3O_4$ nanocomposite was absent. Therefore, the main type of interaction between the rGO and the $Co_3O_4$ nanocube was determined by van der Waals forces. The process was exothermic, and the heat of formation was:

$$\Delta H \ (G \rightarrow GO) = E(GO) - E(G) - 17 \times E(O) = -51.14 \text{ eV}. \tag{5}$$

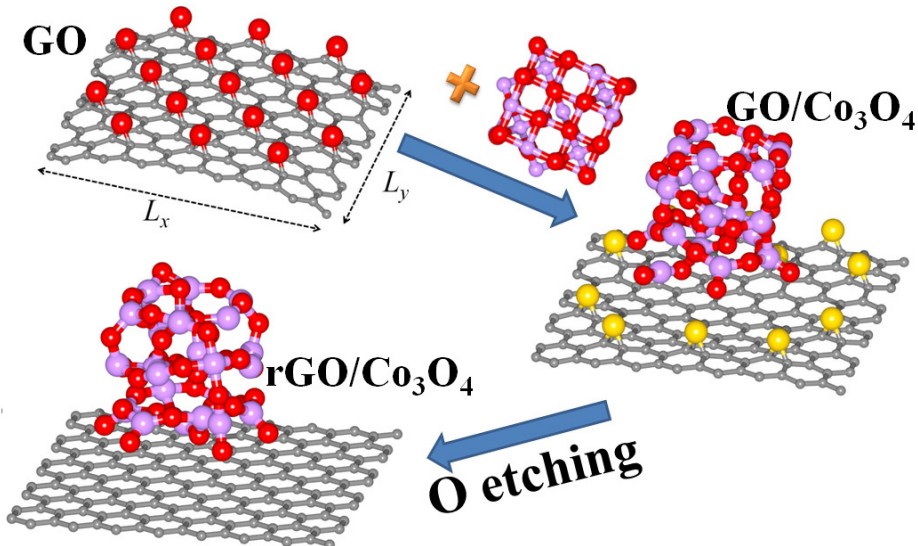

**Figure 2.** The process of the $rGO/Co_3O_4$ atomic supercell building. In the first stage, the $Co_3O_4$ supercell was attached to the surface of the GO cell, then the O atoms that did not bind to the $Co_3O_4$ nanocube (marked by yellow) were removed.

Then, the $Co_3O_4$ nanocube was attached to the obtained GO. After optimization, seven oxygen atoms from the GO surface formed covalent bonds with the $Co_3O_4$ nanocube and the energy of:

$$\Delta H \ (GO \rightarrow GO/Co_3O_4) = E(GO/Co_3O_4) - E(GO) - E(Co_3O_4) = -52.49 \text{ eV} \tag{6}$$

was released. After that, we removed 10 O-atoms from the GO surface that did not bind to the nanocube (these atoms are marked by yellow in Figure 2). This process required additional energy of:

$$\Delta H \ (GO/Co_3O_4 \rightarrow rGO/Co_3O_4) = E(rGO/Co_3O_4) - E(GO/Co_3O_4) + 10 \times E(O) = 32.71 \text{ eV} \tag{7}$$

that indicated an endothermic process. Finally, we calculated the heat of formation of rGO cell from GO:

$$\Delta H \ (GO \rightarrow rGO/Co_3O_4) = \Delta H \ (GO \rightarrow GO/Co_3O_4) + \Delta H \ (GO/Co_3O_4 \rightarrow rGO/Co_3O_4) = -19.78 \text{ eV}. \tag{8}$$

Therefore it is seen that such a process is also exothermic, but it is much more favorable than the formation of the $rGO/Co_3O_4$ nanocomposite ($\Delta H \ (G \rightarrow G/Co_3O_4) = -7.37$ eV). That is why in most experiments, the presence of GO is essential for the synthesis of the $G/Co_3O_4$ nanocomposite.

The same procedure was performed to obtain the energy properties of layered nanocomposites. For this goal, the additional vector of translation (Lz) was added to the periodic box (Figure 3). The optimal Lz in the case of $G/Co_3O_4$ was 11.9 Å, in the case of $G^*/Co_3O_4$ it

was 11.8 Å, and in the case of rGO/$Co_3O_4$ it was 14.5 Å. Note that during incorporation, the Co3o4 nanocube interacted with two graphene oxide layers, so it formed chemical bonds with 15 O atoms. It is notable that in the case of 3D G*/$Co_3O_4$, the rotation of the nanocube was observed. The incorporation of $Co_3O_4$ between G, G* and rGO released −2.76, −6.46 and −7.13 eV, correspondingly. Therefore, it can be concluded that the processes of 3D G/$Co_3O_4$, G*/$Co_3O_4$ and rGO/$Co_3O_4$ and nanocomposite synthesis are less favorable than in the case of 2D nanofilms.

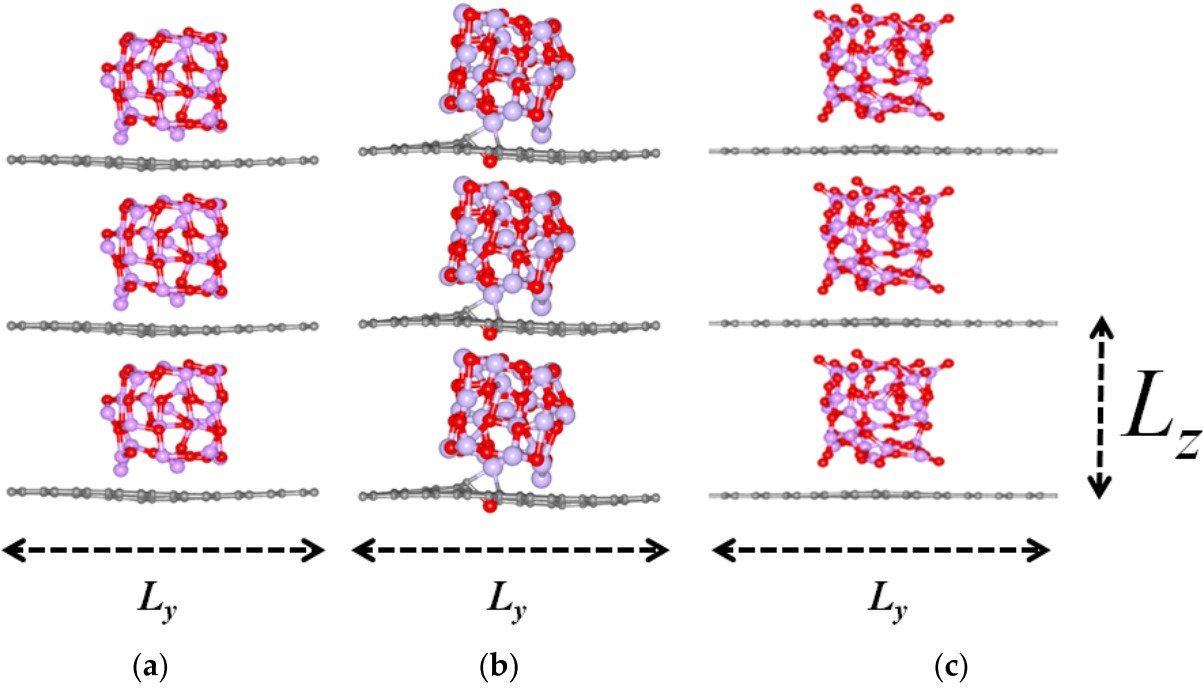

**Figure 3.** The expanded atomic supercells of 3D nanocomposites (**a**) G/$Co_3O_4$ with translations vectors Ly = 21.41 Å, Lz = 11.90 Å; (**b**) G*/$Co_3O_4$ with translations vectors Ly = 21.41 Å, Lz = 11.80 Å; (**c**) rGO/$Co_3O_4$ with translations vectors Ly = 21.62 Å, Lz = 14.50 Å.

*3.2. Capacitive and Electronic Properties*

The plots of QC for the considered structures are shown in Figure 4. The incorporation of in-plane oxygen into graphene carcasses significantly changed the QC near 0 V and created the peak at 0.2 V. The peak follows from the DOS of the structure in which the peak at −4.55 eV is mainly contributed by carbon's p-orbital and slightly by oxygen's p-orbitals (Figures S3 and S4). As it was predicted, the addition of the $Co_3O_4$ nanocube significantly (≈30–40 times) raised the level of graphene's QC, especially at U = 0 V (Table 1). The large peaks of QC were observed near −1 V, which indicated the major contribution of QC to the total capacitance at this value during the charge cycle. We also note that the QC for GO/$Co_3O_4$ and rGO/$Co_3O_4$ composites are almost similar, especially at the level of 0 V. However, the QC of G/$Co_3O_4$ surpasses the QC of GO/$Co_3O_4$ and rGO/$Co_3O_4$ at ≈190 F/g in the case of 2D films (Figure 4a) and at ≈210 F/g in the case of 3D structures. At −1 V, the differences in QC reached ≈300 F/g in the case of 2D films and ≈500 F/g in the case of 3D structures. The QC plot of G*/$Co_3O_4$ almost repeated the QC of G/ $Co_3O_4$ except for the range 0 < U < 0.6 V, where the QC of $Co_3O_4$ is still higher.

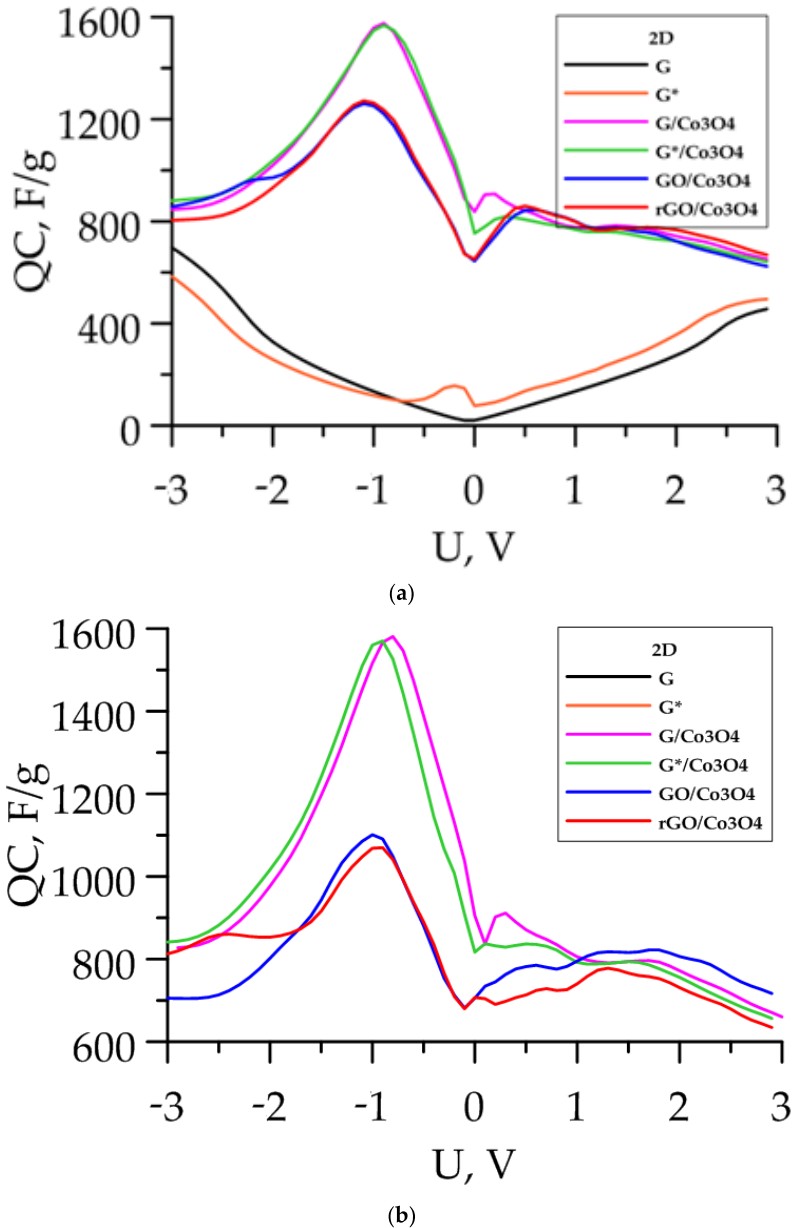

**Figure 4.** The QC for G, G*, G/Co$_3$O$_4$, G*/Co$_3$O$_4$, GO/Co$_3$O$_4$ and rGO/Co$_3$O$_4$: (**a**) 2D films; (**b**) 3D composites.

**Table 1.** Fermi level, charge transfer and the value of QC at the U = 0 V for pure graphene and 2D and 3D composites: G/Co$_3$O$_4$, GO/Co$_3$O$_4$, rGO/Co$_3$O$_4$.

| Structure | Fermi Level, eV | Charge, e | QC (0 V), F/g |
|:---:|:---:|:---:|:---:|
| G | −4.67 | | 20.78 |
| G* | −4.41 | | 111.5 |
| 2D G/Co$_3$O$_4$ | −3.50 | 1.15 | 837.55 |
| 2D G*/Co$_3$O$_4$ | −3.57 | 1.32 | 832.50 |
| 2D GO/Co$_3$O$_4$ | −3.57 | 1.38 | 644.25 |
| 2D rGO/Co$_3$O$_4$ | −3.48 | 0.88 | 652.17 |
| 3D G/Co$_3$O$_4$ | −3.41 | 1.73 | 880.34 |
| 3D G*/Co$_3$O$_4$ | −3.40 | 2.25 | 862.81 |
| 3D GO/Co$_3$O$_4$ | −3.65 | 1.99 | 692.13 |
| 3D rGO/Co$_3$O$_4$ | −3.26 | 1.13 | 691.13 |

Table 1 also lists the Fermi level of the considered structures. The substitution of the C atom by the O atom shifted the Fermi level of pure graphene from $-4.67$ to $-4.41$ eV. The addition of the $Co_3O_4$ nanocube raises the Fermi level to $-3.50$ eV for 2D $G/Co_3O_4$, to $-3.57$ eV for 2D $G^*/Co_3O_4$ and 2D $G^*/Co_3O_4$ and to $-3.48$ for 2D $rGO/Co_3O_4$. Further, it can be seen that in the case of the 3D structure, the Fermi level was less, which indicated the enhancement of its conductive properties in comparison to 2D films. Another interesting fact that confirms the advantage of sandwich-like structures of SCs application is the amount of charge that flowed to graphene layers. In the case of 2D G and rGO particles, this charge was equal to 1.15 and 0.88 e, correspondingly, while in the case of 3D structures, the amount of reported charge enlarges to 1.99 and 1.13 e. The most noticeable charge transfer was observed in the case of 3D $G^*/Co_3O_4$ and was equal to 2.25 e. It can be explained by the formation of the two chemical bonds between graphene and the nanocube.

**4. Conclusions and Discussion**

The attempt to compare adhesion and electronic properties of $G/Co_3O_4$, $G^*/Co_3O_4$ and $rGO/Co_3O_4$ nanocomposites was performed for the first time by the SCC DFTB method. Our calculation showed that the formation of the $rGO/Co_3O_4$ started from the attachment of $Co_3O_4$ nanocubes to the GO surface with consequent etching of oxygen atoms is more favorable than the straight decoration of G and G* (graphene with substitutional oxygen) with $Co_3O_4$ nanoparticles both for 2D films and 3D composites. The addition of $Co_3O_4$ nanoparticles to G shifts the Fermi level to 1.10–1.41 eV, which indicates the enhancement of the conductive properties. The maximum charge transfer that was given by $Co_3O_4$ nanocubes was observed in the case of 3D $G^*/Co_3O_4$ and was equal to 2.25 e. The dependence of QC on voltage showed a significant improvement of capacitance in the case of G and G* in comparison to rGO. Independent of the applied voltage, this difference varies in the range of 300–500 F/g. In the Introduction, it was mentioned that $G/Co_3O_4$ is obtained from GO that leads to the retention of oxygen atoms and to a decrease in total capacitance. Therefore, one of the main challenges in this field is connected to the straight synthesis of $G/Co_3O_4$ from pure G. One of the possible methods is the two-step method in which graphene was initially deposited on a Ni foam with $Co_3O_4$ reported by Zhang [32]. Another variant to obtain materials with high capacity is connected with $G^*/Co_3O_4$ material, the capacitive properties of which are a little less than $G/Co_3O_4$. The high QC, along with its small size, may make the $G/Co_3O_4$ nanocomposite an important element of lightweight and flexible supercapacitors for future portable and wearable devices.

**Supplementary Materials:** The following supporting information can be downloaded at: https://www.mdpi.com/article/10.3390/lubricants10050079/s1, Figure S1: The atomic structure of graphene with substitutional graphitic oxygen (G*). The oxygen is located in one plane with a graphene carcass. There are two C–O bond lengths of 1.47 Å (blue line), and one C–O bond length of 1.48 Å. (green line). The C–C bond length near O atoms decreases to 1.40 Å (orange); Figure S2: The atomic structure of graphene with substitutional graphitic oxygen after attachment to the $Co_3O_4$ nanocube. It is seen that one of the C–O bonds was broken; Figure S3: The density of state (DOS) for pure graphene (G) and graphene with substitutional graphitic oxygen (G*). Dashed lines mark the Fermi level; Figure S4: The total density of state (DOS) for graphene with substitutional graphitic oxygen (G*) and partial DOS (pDOS) for p-orbitals of carbon and oxygen atoms. Dashed line marks the Fermi level.

**Author Contributions:** Conceptualization, V.S. and O.G.; methodology, V.S.; software, O.G.; validation, V.S. and O.G.; formal analysis, O.G.; investigation, V.S.; resources, O.G.; data curation, O.G.; writing—original draft preparation, V.S.; writing—review and editing, V.S. and O.G.; visualization, V.S.; supervision, O.G.; project administration, O.G.; funding acquisition, O.G. All authors have read and agreed to the published version of the manuscript.

**Funding:** This research was funded by the Ministry of Science and Higher Education of the Russian Federation (project no. FSRR-2020-0004). V. V. Shunaev wishes to thank Russian President scholarship, project no. SP-3976.2021.1.

**Institutional Review Board Statement:** Not applicable.

**Informed Consent Statement:** Not applicable.

**Data Availability Statement:** The data presented in this study are available on request from the corresponding author.

**Conflicts of Interest:** The authors declare no conflict of interest.

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
