# Peer review of "Interaction of Co3O4 Nanocube with Graphene and Reduced Graphene Oxide: Adhesion and Quantum Capacitance"

_lubricants, doi:10.3390/lubricants10050079_

Round 1
Reviewer 1 Report
Comments to the Authors
The manuscript reports the interaction of Co3O4 nanocube with graphene and reduced graphene oxide based on the SCC DFTB method. The results show that the formation of the rGO/Co3O4 started from attachment of Co3O4 nanocubes to GO surface with consequent etching of oxygen atoms is more favorable than the straight decoration of pure G with Co3O4 nanoparticles both for 2D-films and 3D-composites. The calculations are performed properly and the manuscript is well written. The manuscript is suggested for publication after solving the following small issues,
- In the ‘Methods’ part (line 77), the authors wrote “All calculations were performed at 300 K…”. I think the calculations were performed at 0 K instead of 300 K. The 300 K is probably the electron temperature, which is for the smearing but not real temperature. The authors should check this.
- In the manuscript, the character ‘÷’ appears several times. For example, in the abstract part (line 16), “in the range of 300÷500 F/g…”. At line 162, “–50÷–3.32 eV.”. At line 175, “1.10÷1.41 eV…”. I cannot understand the meaning of ‘÷’.
Author Response
The manuscript reports the interaction of Co3O4 nanocube with graphene and reduced graphene oxide based on the SCC DFTB method. The results show that the formation of the rGO/Co3O4 started from attachment of Co3O4 nanocubes to GO surface with consequent etching of oxygen atoms is more favorable than the straight decoration of pure G with Co3O4 nanoparticles both for 2D-films and 3D-composites. The calculations are performed properly and the manuscript is well written. The manuscript is suggested for publication after solving the following small issues,
Thank you very much!
- In the ‘Methods’ part (line 77), the authors wrote “All calculations were performed at 300 K…”. I think the calculations were performed at 0 K instead of 300 K. The 300 K is probably the electron temperature, which is for the smearing but not real temperature. The authors should check this.
We thank Reviewer for this important notice. Reviewer is absolutely right: 300 К is electron temperature. The real temperature which stands for atoms velocities is 0K. We fixed in the text.
- In the manuscript, the character ‘÷’ appears several times. For example, in the abstract part (line 16), “in the range of 300÷500 F/g…”. At line 162, “–50÷–3.32 eV.”. At line 175, “1.10÷1.41 eV…”. I cannot understand the meaning of ‘÷’.
Under symbol «÷» we meant the range. For example, «capacitance in the range of 300÷500 F/g» means that capacitance changes from 300 to 500 F/g. We will note the attention of Editors to this symbol. If it is not widely used it can be corrected.
Reviewer 2 Report
The manuscript "Interaction of Co3O4 Nanocube with Graphene and Reduced Graphene Oxide : Adhesion and Quantum Capacitance" by Shunaev et al. is on the topic of the realization of efficient supercapacitors and batteries by using transition metal oxides and graphene oxide (GO) / reduced graphene oxide (rGO).
The subject matter of this work and the extracted conclusions are interesting. Authors show detailed (first principles) and realistic calculations considering the interaction of the Co3O4 nanocubes with GO,rGO. Therefore, these calculations are close to experimental conditions and well beyond state-of-the-art theoretical works only considering the interaction between Co3O4 particles and pristine graphene. Furthermore, results are clear and well described and the manuscript is well-written. I believe this manuscript is suitable for publication if the following issues are addressed:
1. I believe the abstract should be clearer. Initial sentences in this section refer to experiments (synthesis of Co3O4 and graphene composites). Then, authors refer to their quantum chemical calculations, and thus the reader may not know at this point whether the article is focused on experiments or theory. I would advice authors to include a clarifying and joining sentence stating that they theoretically study the synthesis of these composites via first principle calculations.
2. The conclusion that GO/Co3O4 and rGO/Co3O4 composites have a smaller capacitance than G/Co3O4 is very interesting and relevant for the experimental design of these advanced nanomaterials. As shown in Fig.2, this conclusion is reached by noticing that, in experimental GO and rGO, oxygen atoms are present at the surface of graphene. I also point out that recent studies have demonstrated that GO and rGO also present substitutional (i.e. in plane) oxygen in graphene [Hofer et al. Nature Communications, 10, 4570, 2019; Mackenzie et al., 2D Materials, 8, 045035, 2021]. Moreover, the electronic properties of graphene with in-plane oxygen atoms are notably different to those of graphene with oxygen-containing functional groups bound to the basal plane [Mackenzie et al., 2D Materials, 8, 045035, 2021]. Substitutional oxygen in the graphene lattice could, therefore, modify the overall calculated capacitance of the nanocomposite.
I recommend authors to calculate the adhesion and capacitance of the composite formed by graphene with in-plane oxygen / Co3O4. One of the many existing configurations (e.g. graphitic substitution) would be sufficient. Such calculations would notably strengthen the relevance of the overall study.
Author Response
We thak Reviewer for his kind comment!
The manuscript "Interaction of Co3O4 Nanocube with Graphene and Reduced Graphene Oxide : Adhesion and Quantum Capacitance" by Shunaev et al. is on the topic of the realization of efficient supercapacitors and batteries by using transition metal oxides and graphene oxide (GO) / reduced graphene oxide (rGO).
The subject matter of this work and the extracted conclusions are interesting. Authors show detailed (first principles) and realistic calculations considering the interaction of the Co3O4 nanocubes with GO,rGO. Therefore, these calculations are close to experimental conditions and well beyond state-of-the-art theoretical works only considering the interaction between Co3O4 particles and pristine graphene. Furthermore, results are clear and well described and the manuscript is well-written.
Thank you very much!
I believe this manuscript is suitable for publication if the following issues are addressed:
- I believe the abstract should be clearer. Initial sentences in this section refer to experiments (synthesis of Co3O4 and graphene composites). Then, authors refer to their quantum chemical calculations, and thus the reader may not know at this point whether the article is focused on experiments or theory. I would advice authors to include a clarifying and joining sentence stating that they theoretically study the synthesis of these composites via first principle calculations.
Thanks for this remark. We add sentence «In this article we will make an effort to solve this problem theoretically» to the Abstract. Also the name of the method (SCC DFTB) is mentioned in the end of Abstract.
- The conclusion that GO/Co3O4 and rGO/Co3O4 composites have a smaller capacitance than G/Co3O4 is very interesting and relevant for the experimental design of these advanced nanomaterials. As shown in Fig.2, this conclusion is reached by noticing that, in experimental GO and rGO, oxygen atoms are present at the surface of graphene. I also point out that recent studies have demonstrated that GO and rGO also present substitutional (i.e. in plane) oxygen in graphene [Hofer et al. Nature Communications, 10, 4570, 2019; Mackenzie et al., 2D Materials, 8, 045035, 2021]. Moreover, the electronic properties of graphene with in-plane oxygen atoms are notably different to those of graphene with oxygen-containing functional groups bound to the basal plane [Mackenzie et al., 2D Materials, 8, 045035, 2021]. Substitutional oxygen in the graphene lattice could, therefore, modify the overall calculated capacitance of the nanocomposite.
I recommend authors to calculate the adhesion and capacitance of the composite formed by graphene with in-plane oxygen / Co3O4. One of the many existing configurations (e.g. graphitic substitution) would be sufficient. Such calculations would notably strengthen the relevance of the overall study.
Thank you for interesting suggestion. As we see, in Hofer et al. Nature Communications, 10, 4570, 2019 major part of in-plane oxygen is formed in holes of graphene (except graphitic substitution) so here we can say that the object of study is graphene nanomesh (or holey graphene) with little holes. Atoms in its hole have unpassivated active dangling bonds that makes possible to absorb different atoms. We performed calculations with decoration of graphene nanomesh by various doping (https://doi.org/10.1002/admt.202101250; https://doi.org/10.22226/2410-3535-2021-4-392-396) and we are sure that such comprehensive modeling requires a lot of time (much more than 10 days needed for review’s answer). An graphene nanomesh also is another object of study.
Another important moment that the goal of our work is to find out the most energy favorable way of G/Co3O4 synthesis. We showed that attachment of Co3O4 nanocubes to GO with subsequent etching of oxygen is much more favorable than attachment of Co3O4 particles to pure graphene. In suggested by Reviewer Hofer et al. Nature Communications, 10, 4570, 2019 graphene with in-plane oxygen atoms was received after preliminary electron irridation of graphene that lead to defects in its plane; in Mackenzie et al., 2D Materials, 8, 045035, 2021 the O implantation into graphene was realized via a controlled plasma-based process. So we see that obtaining of such structure as graphene with in-plane oxygen requires much more energy-consuming efforts that considered in our article.
Round 2
Reviewer 2 Report
This document is a revision of a manuscript I initially reviewed. My report included two key points (requests for clarification, and technical remarks) to be addressed. I’ve now read the authors’ rebuttal letter to each of the points and the new version of the manuscript.
I find that the new manuscript still suffers from the reported issues (see details below). As such, I cannot recommend this article for publication.
- As argued in my previous report, the abstract should clearly state the fact that the study is theoretical. Otherwise, readers may be confused while reading it. Despite requested, authors have not added any clarifying sentence in the new version of the abstract. Instead, authors have included a sentence in the last paragraph of the introductory paragraph, but this important information should appear at an earlier stage in the manuscript.
- In my previous report, I have recommended authors to calculate the adhesion and capacitance of the composite formed by graphene with in-plane oxygen / Co3O4. The reason is clear and justified: recent studies have demonstrated that, apart from out-of-plane oxygen functional groups, GO and rGO also present substitutional (i.e. in plane) oxygen in graphene [Hofer et al. Nature Communications, 10, 4570, 2019]. More importantly, the electronic properties of graphene with in-plane oxygen are radically different from those of graphene with oxygen-containing functional groups on the basal plane [Mackenzie et al., 2D Materials, 8, 045035, 2021], and this may alter the calculated capacitance of the composite. Nonetheless, authors have not performed this pertinent calculation.
Here, I further justify why this calculation is relevant for the present study. These explanations also serve to address the comments stated by authors in the rebuttal letter.
- As demonstrated in [Hofer et al. Nature Communications, 10, 4570, 2019], conventional chemical (oxidation) methods to prepare GO and rGO (see Methods section) lead to the insertion of in-plane oxygen in graphene. In other words, there is no need to irradiate graphene to observe these in-plane impurities. For clarity, I note that authors in [Hofer et al. Nature Communications, 10, 4570, 2019] only assume that functional groups attached to graphene´s basal plane via weaker bonds are destroyed before a TEM image can be obtained due to the electron irradiation. However, in-plane oxygen in the graphene lattice is still present and can be imaged after such irradiation, meaning that these configurations are indeed very stable and thus relevant.
- Configurations where oxygen substitutes a carbon atom in the graphene lattice are quite probable, even those that do not involve the creation of a vacancy. This is clearly demonstrated in Figure 2e in [Hofer et al. Nature Communications, 10, 4570, 2019], which shows the distribution of the different configurations of oxygen in GO. In particular, the configurations “oxygen pair” and “graphitic oxygen” are two configurations where oxygen directly substitutes carbon atoms in the graphene lattice and constitute ~50% of the total relative counts. I emphasize that none of these two configurations are associated with the creation of any vacancy/holes in the graphene lattice, and therefore their simulation is easier than configurations with vacancies / dangling bonds.
Due to the former reasons, I encourage authors to calculate the adhesion and capacitance of the composite formed by graphene with in-plane oxygen / Co3O4 (one configuration would be sufficient, I recommended “graphitic oxygen” for simplicity but “oxygen pair” would also serve). This calculation would notably strengthen the relevance of their study.
Author Response
Dear Reviewer! We improved tha article according to you notes. We greatly thank you for valuable advices !
- As argued in my previous report, the abstract should clearly state the fact that the study is theoretical. Otherwise, readers may be confused while reading it. Despite requested, authors have not added any clarifying sentence in the new version of the abstract. Instead, authors have included a sentence in the last paragraph of the introductory paragraph, but this important information should appear at an earlier stage in the manuscript.
In the Abstract it was written that we perfromed "quantum chemical calculations ". Now we added the word "theoretical". I think that that potentional readers will not be confused.
2. Due to the former reasons, I encourage authors to calculate the adhesion and capacitance of the composite formed by graphene with in-plane oxygen / Co3O4 (one configuration would be sufficient, I recommended “graphitic oxygen” for simplicity but “oxygen pair” would also serve). This calculation would notably strengthen the relevance of their study.
We performed extensive additional calculation for graphene with in plane graphitic oxygen. The new data was added to the paper and were highlighted by blue. Several interesting results were obtained for the first time. It's seen that capacitive properties of Co3O4 on graphene with substitutional oxygen are lower that for pure graphene but higher than for rGO.
WB
Authors
Round 3
Reviewer 2 Report
Authors have satisfactorily addressed the suggested comments. Several interesting conclusions can be drawn from the presented calculations, which makes the study highly relevant in the field of nanomaterials for energy storage. I strongly recommend the publication of the new version of the manuscript.
Author Response
Thank you very much for your time that you gave for improving of our paper!
Best regards, Authors